# MICROENVIRONMENT PROBABILITY FLOWS AS PROFICIENT PROTEIN ENGINEERS

## ABSTRACT

The inverse folding of proteins has tremendous applications in protein design and protein engineering. While machine learning approaches for inverse folding have made significant advancements in recent years, efficient generation of diverse and high-quality sequences remains a significant challenge, limiting their practical utility in protein design and engineering. We propose a probabilistic flow framework that introduces three key designs for designing an amino acid sequence with target fold. 1) At the input level, compare to existing inverse folding methods, rather than sampling sequences from the backbone scaffold, we demonstrate that analyzing a protein structure via the local chemical environment (microenvironment) at each residue can come to comparable performance. 2) At the method level, rather than optimizing the recovery ratio, we generate diverse suggestions. 3) At the data level, during training, we propose to do data augmentation with sequence with high sequence similarity, and train a probability flow model to capture the diverse sequence information. We demonstrate that we achieve comparable recovery ratio as the SOTA inverse folding models with higher inference efficiency and flexibility by only using micro-environment as inputs, and further show that we outperforms existing inverse folding methods in several zero-shot thermal stability change prediction tasks.

## 1 INTRODUCTION

Protein engineering and design (*e.g.*, AlQuraishi, 2019; Kuhlman et al., 2003; Huang et al., 2016; Kuhlman & Bradley, 2019) are rapidly growing interdisciplinary fields that encompass computational and experimental methods aimed at discovering amino acid sequences to achieve desired functions or physicochemical properties (*e.g.*, Kuhlman & Bradley, 2019; Shroff et al., 2020; Paik et al., 2021; Yang et al., 2019; Wittmann et al., 2021). One of the key challenges in protein engineering is identifying residue primed to improve a particular phenotype such as expression (Daly & Hearn, 2005), stability (Socha & Tokuriki, 2013), activity (Fox et al., 2003), etc, upon mutation. Machine learning (ML)-guided protein engineering has emerged as a promising approach to address this challenge; so far, several machine learning algorithms have demonstrated the ability to learn meaningful representations of the mutational landscape to accelerate the identification of gain-of-function mutations. (*e.g.*, Shroff et al., 2020; Paik et al., 2021; Yang et al., 2019; Wittmann et al., 2021; d'Oelsnitz et al., 2023; Lu et al., 2022b; Diaz et al., 2023).

One common practice in structure-based ML-guided protein design and engineering is to train on self-supervised learning tasks, such as inverse folding, and masked residue prediction, to learn per-residue likelihoods of amino acids. (*e.g.*, Shroff et al., 2020; Lu et al., 2022b; Paik et al., 2021; d'Oelsnitz et al., 2023). Inverse folding aims to predict the amino acid sequence that can fold into a specific protein backbone scaffold while masked residue prediction aims to predict the masked amino acid from its surrounding atomistic chemical environment (microenvironment) (Dauparas et al., 2022; Torng & Altman, 2017). These self-supervised tasks enable scaling to vast amounts of unlabeled protein data and provide meaningful representations to fine-tune on downstream tasks (Townshend et al., 2020; Jing et al., 2020).

It is quite common in nature for proteins that have a sequence similarity of at least ∼30% to have a structurally similar scaffold (Stern, 2013; Rost, 1999). Once we go below 30% sequence similarity, this fact begins to break down and is known as the "twilight zone" (Rost, 1999). This scaffold

degeneracy is the underlying principle that drives the grouping of proteins into families and domains (*e.g.*, Fox et al., 2014; Sillitoe et al., 2021; Paysan-Lafosse et al., 2023). However, current inverse-folding frameworks demonstrate a narrow range of sequence diversity compared to the observed distribution of extant proteins. This has motivated the development of ML methods that can suggest a diverse set of functionally plausible sequences for a given fold (Li et al., 2020). Here, we propose one micro-environment based probability flow (MeFlow): a rectified flow-based method (Liu, 2022; Lipman et al., 2022) that predicts protein sequences by representing the protein backbone as per-residue micro-environments. To promote more diverse generation, we augment the wild-type protein sequences using the other sequences that exhibit high sequence similarity.

To assess the effectiveness of our approach for protein sequence prediction, we compare the wild-type sequence recovery ratio with other methods on CATH 4.2 (Sillitoe et al., 2021). CATH is a widely used database that categorizes proteins based on their evolutionary structural similarity. We achieve high recovery ratios, indicating its effectiveness in accurately predicting a scaffold's native sequence. To further evaluate the folding quality of the generated diverse sequences, we compute the AlphaFoldV2 (Jumper et al., 2021) scores. To evaluate diversity, we compare relative hamming distance and ESM2 embedding distance of the generated sequences versus wildtype and show MeFlow produces a more diverse set of sequences.

**Contributions** We present three key contributions: (1) we use a masked micro-environment approach to perform amino acid prediction for target fold, which leads to faster training and inference and improved accuracy; (2) we give the first probability flow based method to generate diverse sequences conditioned on micro-environments with data augmentation, leading to considerably more diverse overall sequences versus state-of-the-art inverse folding tools; (3) we illustrate that top-performing inverse folding tools might not be as effective in zero-shot single-mutation engineering scenarios. More specifically:

- MeFlow splits the protein scaffold into per-residue microenvironments instead of using the entire backbone scaffold as the input. This makes it easy to parallelize, leading to inference time on the order of seconds while training time is approximately five GPU days. the recovery ratio aligns with or even outperforms the ones with the entire backbone structure as inputs. In comparison, MPNN (Dauparas et al., 2022) inference takes over 1,000 seconds and ESM inverse folding (ESM-IF) (Hsu et al., 2022) takes hundreds of GPU days to train. Moreover, MeFlow achieves a $53.53\%$ recovery ratio on CATH; while MPNN and ESM-IF achieves lower than $50\%$ recovery ratio.

- MeFlow yields the first probability flow generative model for sequences conditioned on a particular micro-environment. We further introduce data augmentation in the amino acid label space to encourage diversity. This provides faster and more expressive exploration of the solution space compared to prior works, (e.g., beam-search with MPNN (Dauparas et al., 2022)). It improves sequence diversity (as measured by traditional metrics such as hamming distance and ESM2-embedding distance) by $10\%$ over prior work.

- On several single-point mutation fitness prediction benchmarks, our model achieves improvements over ProteinMPNN. In contrast, despite their recent advancements, current high-accuracy inverse folding tools still lag behind ProteinMPNN. Our analysis underscores the significance of incorporating diversity during model design.

## 2 BACKGROUND AND RELATED WORKS

**Machine Learning Based Inverse Folding**  Inverse folding is the problem of designing a protein sequence that will fold into a target scaffold with desired properties (Godzik et al., 1993). It has been a long-standing challenge in computational biology and has important applications in protein-based biotechnology. Physics-based models have also been developed for inverse folding (*e.g.*, Alford et al., 2017; Dahiyat & Mayo, 1997; DeGrado, 1997). These methods use energy functions which directly models the physical basis of a protein's folded state and search for the optimal sequence that maximizes the thermodynamic stability of a given protein structure. Recently, machine learning approaches have shown promising results in addressing the inverse folding problem (*e.g.*, Hsu et al., 2022; Dauparas et al., 2022; Gao et al., 2022).

The most effective approach is to train an auto-regressive prediction model that generates amino acid sequences. For example, ESM-IF (Hsu et al., 2022) uses an encoder-decoder architecture that makes use of geometric vector perceptron (GVP) and transformer blocks and outputs the most possible amino acid sequence given the backbone structure as input. ProteinMPNN (Dauparas et al., 2022) turns the protein backbone into a residue-level graph and uses a message-passing graph neural network (MPNN) in an encoder-decoder architecture to decode the sequence in a sequential or non-sequential auto-regressive fashion. PiFold (Gao et al., 2022) introduces non-autoregressive fashion and GVF (Mao et al., 2023) improves the GVP layer of a GNN, and come to better recovery ratio in commonly-used benchmarks. Overall, machine learning-based methods for inverse folding are a valuable approach to protein design because they can outperform physic-based methods with a fraction of the computational cost. Zheng et al. (2023) proposes to use inverse folding model information to improve a masked language model based protein designer.

Another possible approach is to use recently-developed probability flow based models. In the literature, researchers have used generative models to design small molecules or DNA sequences (*e.g.*, Brock et al., 2016; Wu et al., 2022; Trippe et al., 2022; Gupta & Zou, 2019; Xu et al., 2022). In our work, we try to solve the problem only with micro-environment information to understand whether it is necessary to model the full backbone as inputs. We further apply probabilistic flow (e.g., diffusion) model and display the advantages for single-mutation protein engineering.

**Probabilistic Flows**   In recent years, researchers have been trying to improve the generative models by breaking down the one-step mapping into multiple steps (*e.g.*, Song & Ermon, 2019; Song et al., 2020; Song & Ermon, 2020; Song et al., 2021; Zhang et al., 2023). This can be done in an ODE (Zhang et al., 2022; Liu, 2022) or SDE (Meng et al., 2021; Song et al., 2020) fashion. The denoising diffusion probabilistic models (DDPM) Ho et al. (2020) are among the approaches that have demonstrated impressive flexibility and power in generating high-quality samples in various domains, such as large-scale image benchmarks (*e.g.*, Ho & Salimans, 2022; Saharia et al., 2022; Nichol et al., 2021). Consequently, the diffusion model has become a mainstream approach to transporting noise to the target distribution. Despite the success of DDPM in generating high-quality samples on large-scale benchmarks, a drawback with this approach is that it requires hundreds of simulation steps to generate desired samples. Previous works have proposed strategies to reduce the simulation steps and accelerate the learning of the transport process. For instance, DDIM (Zhang et al., 2022) formulates the sampling trajectory process as an ODE. FastDPM (Kong & Ping, 2021) and its variants bridge the connection between the discrete and continuous time steps. However, simplifying generation process into a few step simulation remains a challenge; knowledge distillation methods have shown promise but have yet to match the performance with a single or few steps. Recently, (Liu, 2022; Liu et al., 2022) propose *Rectified Flow* and uses *ReFlow* operation to reduce the inference simulation steps into a single step.

Neural probabilistic flows such as Neural bridges (Wu et al., 2022; Wang et al., 2021) and DDPM, have recently demonstrated their powers in image generation and other real-world applications (Nichol et al., 2021; Lu et al., 2022a). These models repeatedly feed forward the input data dependent on time $t \in [0, 1]$ and then output the final output. The core idea is to learn a probabilistic model that can generate samples from a target distribution $\pi_1$ by gradually transporting a distribution $\pi_0$ through a sequence of diffusion steps. Therefore, during training, these models target mimicking trajectories from the real distribution to a random distribution. For example, the well-known method DDPM (Ho et al., 2020) trains the model with

$$\min_\theta \mathbb{E}_{t, x_1, \epsilon} \left[ \| \epsilon - v_\theta(\sqrt{\hat{\alpha}_t} x_1 + \sqrt{1 - \hat{\alpha}_t} \epsilon, t) \|^2 \right], \tag{1}$$

where $v_\theta$ is the model parameterized by $\theta$, $x_1$ denotes the real data and $\epsilon = x_0$ is the random noise. Intuitively, $\sqrt{\hat{\alpha}_t} x_1 + \sqrt{1 - \hat{\alpha}_t} \epsilon$ is the interpolation of $x_1 \sim \pi_1$ and $\epsilon \sim \pi_0$. Recently, Liu (2022) proposes to simplify the training objective by converting the trajectories to straight lines and removing the noise on the trajectories. Concurrent works (Lipman et al., 2022; Liu et al., 2022) analyze more variants and theoretical results, and we introduce more details in the Section 3.

## 3 METHOD

**Local Chemical Environments as Inputs** We aim to identify the amino acid sequence that aligns with a specific protein backbone structure. In this study, we diverge from the traditional full backbone structure approach to inverse folding. Instead, we focus solely on the local chemical environment as our input. Consequently, for a single protein sequence's backbone, we partition it into $M$ distinct micro-environments, each corresponding to an amino acid in the sequence. Our goal is to explore whether the overarching global structure or the intricate local structure has difference in achieving the desired target fold.

Consider a local chemical environment within a protein's backbone structure. This environment is represented by a set of atoms, $\text{Env} = \{a_i\}_{i=1}^{N}$ where each atom $a_i = \{c_i, o_i, p_i\}$ comprises its atom type $o_i$, the 3D coordinate $c_i$ and the physical properties $p_i$. Our objective is to determine the target amino acid $X$ given all the atoms $\{a_i\}_{i=1}^{N}$ as inputs. $X \in \mathbb{F}_2^{20}$ is a set of one-hot encoded vectors for the 20 amino acid (side chain) type, $\mathbb{F}_2 = \{0, 1\}$ is a finite space.

**Training Probability Flow Network** We employ the rectified flow (Liu, 2022; Liu et al., 2022) to construct a conditional generative model that generates amino acid types based on the provided micro-environment. Our model starts with a Gaussian noise $X_0 \in \mathbb{R}^{M \times 20}$ at $t = 0$ and uses an ODE to update it to $X_1 \in \mathbb{R}^{M \times 20}$, which matches the data distribution. We relax $X$ to the continuous space $\mathbb{R}$ instead of $\mathbb{F}_2$ and use $v_\theta$ to denote the velocity field network, which is defined by the following process

$$\underbrace{\mathrm{d}X_t}_{\text{drift}} = \underbrace{v_\theta(X_t, t \mid \text{Env})}_{\text{velocity}} \underbrace{\mathrm{d}t}_{\text{time interval}}, \quad \text{with } t \in [0, 1], \tag{2}$$

where $X_t$ is the interpolated amino acid type at time $t$ and the velocity filed $v_\theta$ is a neural network with $\theta$ as its parameters and Env, the local micro-environment, is the conditional information. The optimal direction at any time $t$ is $X_1 - X_0$. Thus, we can encourage our velocity field to directly follow the optimal ODE process $\mathrm{d}X_t = (X_1 - X_0)\mathrm{d}t$ by optimizing

$$\min_\theta \int_0^1 \mathbb{E}\left[\|(v_\theta(X_t, t \mid \text{Env}) - (X_1 - X_0)\|^2\right]\mathrm{d}t,$$
$$\text{where } X_t = tX_1 + (1 - t)X_0 \quad t \in [0, 1]. \tag{3}$$

Empirically, we do not optimize the loss in (3) with the integration on $t \in [0, 1]$ directly. Instead, for each data sample $X_1$, we randomly draw a $X_0$ from random Gaussian noise, a $t$ from $[0, 1]$, and minimize the following loss,

$$\min_\theta \mathbb{E}_{t \sim \mathcal{U}(0,1), (X_1, \text{Env}) \sim \mathcal{D}, X_0 \sim \mathcal{N}(0,1)}\left[\|(v_\theta(X_t, t \mid \text{Env}) - (X_1 - X_0)\|^2\right], \tag{4}$$

where $\mathcal{D}$ denotes the training data. After the neural velocity field $v_\theta$ is well-trained, samples can be generated by discretizing the ODE process with Euler solver in (2) into $N$ steps (*e.g.*, $N = 1000$),

$$X'_{\hat{t}+1/N} \longleftarrow X'_{\hat{t}} + \frac{1}{N} v_\theta(X'_{\hat{t}}, \hat{t}), \tag{5}$$

the time step $\hat{t}$ is defined as $\hat{t} \in [0, 1]$. Here $X'_1$ denotes our generated samples and $X'_0 = X_0$. Intuitively, the Euler solver will be more accurate with a large $N$ and a better solver can come to more accurate results.

**ReFlow for Efficient Inference** As discussed in Liu (2022) and Liu et al. (2022), once we can get a one-step model that can generate data by

$$X'_1 = X'_0 + v_\theta(X'_0, t = 0 \mid \text{Env}), \tag{6}$$

and it will be easy to embed the amino acids into latent space and do editing for the one-step model since we create a one-to-one function. For this purpose, we follow (Liu, 2022; Liu et al., 2022) to use *ReFlow* to refine the neural ODE process learned with $v_\theta$ in the previous stage. We construct the objective as follows,

$$\min_\theta \mathbb{E}\left[\|(v_\theta(X'_t, t \mid \text{Env}) - (X'_1 - X_0)\|^2\right], \quad t \sim \mathcal{U}(0,1), \tag{7}$$

where $X'_1$ is generated from our first model optimized with (4) and $X'_t = tX'_1 + (1 - t)X_0$. In our experiments, we observed that *ReFlow* improves the data quality generated by a single-step model.

**Data Augmentation** To harness the generative models' capability for diverse outputs, we suggest augmenting the wild-type protein sequence with others that have high sequence similarity. Based on the principle that sequence dictates structure, a high sequence similarity, such as 90%, often indicates that two protein sequences might share structural similarities. For every protein sequence in our training dataset, we extract the top-5 sequences from the Uniref100 dataset (sourced on November 24, 2021) (Suzek et al., 2007) using MMseqs2 [1], yielding an average sequence similarity of approximately $\sim 85\%$. Throughout the training process, we augment the ground-truth protein sequence with these 5 additional sequences.

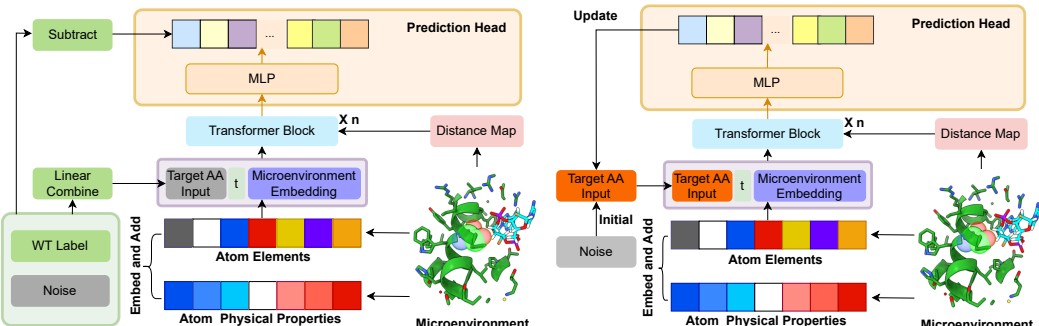

Figure 1: We demonstrate the training and inference framework of our algorithm. **Left.** We display that, during training, we linearly combine the wild-type and noise as input and calculate the loss in (4). the model's input during training is $X_t$, a linear combination of $X_0$ (random noise) and $X_1$ (one-hot label). The subtraction of $X_1$ and $X_0$ symbolizes the drift from $X_0$ to $X_1$. $t$ denotes the time step. **Right.** The right figure shows the inference process, in which we start from random noise $X_0$ and update it with (5). $t$ denotes the time step.

**Model Architecture** Our architecture is based on the graph transformer neural network architecture proposed for self-supervised mask prediction in Diaz et al. (2023). Specifically, we use this architecture to construct $v_\theta$, which takes the local atom-level environment surrounding the target amino acid as input. The neural network takes in the coordinates, atom types, and physical properties of the atoms, and applies an embedding layer to convert the categorical atom types and physical properties into continuous representations. The embeddings of atom types and physical property types are concatenated to form the input features, which then pass through several attention blocks. In each attention block, there are two attention layers and one MLP layer. The attention layers use the atom-wise Euclidean distance to calculate the attention bias. We refer the reader to Appendix A for more details about the architecture.

**Algorithm** We summarize and visualize our method in Figure 1. Generally speaking, we create a conditional generative model, which generates the amino acid types when given the microenvironment. We use the atom type and additional physical properties as inputs and calculate the atom-wise distance as the attention bias information. After we train the model, we use *ReFlow* to reduce the generation step into 1 to make the inference more efficient.

# 4 EXPERIMENTS

In order to evaluate the performance of MeFlow, we conduct a comprehensive set of experiments addressing pivotal queries. ① We examine the accuracy and diversity of our method and compare it to other state-of-the-art methods. We highlight comparison with two baselines, one is a classification model with micro-environment as input, the other is diffusion model with micro-environment as input. ② We train our MeFlow on the *interface* dataset (d'Oelsnitz et al., 2023), and measure the recovery ratio on multi-domain test sets. ③ We carry out zero-shot single-point mutation fitness prediction tests to showcase our model's potential benefits. Our findings suggest that diversity is crucial in this context, and underscore that models with high accuracy don't always translate to optimal design outcomes.

---

[1] https://github.com/soedinglab/MMseqs2

### 4.1 Amino Acid Prediction

**Datasets and Experiment Settings** We first train our method on commonly-used benchmarks and measure the recovery ratio and diversity. We first evaluate on CATH (Pearl et al., 2003) (Class, Architecture, Topology, Homology) database, which is a hierarchical classification of protein domain structures based on their evolutionary relationships, structural and functional features. The CATH 4.2 includes over 20k protein structures. We follow the same data splitting in (Jing et al., 2020) to split the train and test dataset. To measure the quality of generated sequence (or sub-sequence), we report the recovery ratio (accuracy) by averaging the results on three random noise inputs and the AlphaFold scores on multiple suggested sequences. To measure the diversity, we first calculate the hamming distance. We further measure diversity by encoding sequence with ESM2 (Lin et al., 2022) and measure their difference in the latent space.

We generate another micro-environment dataset, *Interface*. For each protein sequence, we sample all residues at the interface (within 5Å) of ligands with at least 3 carbon atoms, nucleotides, halogens, and cations and then randomly sampled to backfill up to 200 residues or half the length of the protein sequence. The intent is to ensure that we sample the limited amount of non-interface data in the PDB while not oversampling large proteins. Finally, we do a 90/10 split and combine to create our training and test datasets. We make a training set with 2.2M micro-environments and generate multiple domains (*e.g.*, metal interface, RNA interface, etc.) test datasets to verify the model performance. We refer the readers to d'Oelsnitz et al. (2023) for more details about the dataset. In general, our method, which utilizes atom-level micro-environments as inputs, offers great flexibility for various use cases. By simply adding or removing information from the micro-environment, our method can easily accommodate different requirements, making it highly versatile. We further elucidate that models trained on this dataset are adept protein engineering tools.

**Hyper-parameter Settings** In all our experiments, we set the number of blocks to 8, the maximum number of atoms to 256, the number of channels to 128, and batch size 256 (256 different micro-environments) with 300K iterations training with (4) and 100K iterations training with (7). On four A100 GPUs, it takes approximately 1.5 days to train the model on different datasets and settings.

| Model | Perplexity | | | Recovery Ratio (%) | | |
|---|---|---|---|---|---|---|
| | Short ↓ | Single-chain ↓ | All ↓ | Short ↑ | Single-chain ↑ | All↑ |
| StructGNN (Ingraham et al., 2019) | 8.29 | 8.74 | 6.40 | 29.44 | 28.26 | 35.91 |
| GCA (Tan et al., 2022) | 7.09 | 7.49 | 6.05 | 32.62 | 31.10 | 37.64 |
| GVP (Jing et al., 2020) | 7.23 | 7.84 | 5.36 | 30.60 | 28.95 | 39.47 |
| AlphaDesign (Tan et al., 2022) | 7.32 | 7.63 | 6.30 | 34.16 | 32.66 | 41.31 |
| ESM-IF* (Hsu et al., 2022) | 8.18 | 6.33 | 6.44 | 31.30 | 38.50 | 38.30 |
| ProteinMPNN (Dauparas et al., 2022) | 6.21 | 6.68 | 4.61 | 36.35 | 34.43 | 45.96 |
| PiFold (Gao et al., 2022) | 6.04 | 6.31 | 4.55 | 39.84 | 38.53 | 51.66 |
| Classifier | **5.85±0.12** | **6.02±0.11** | **4.26±0.09** | **46.41±0.45** | **45.24±0.36** | **55.37±0.48** |
| Diffusion | 6.71±0.42 | 6.82±0.54 | 5.23±0.47 | 40.17±0.43 | 39.58±0.42 | 50.52±0.44 |
| MeFlow | 6.35±0.39 | 6.46±0.48 | 4.72±0.44 | 42.58±0.33 | 42.14±0.35 | 53.53±0.45 |
| MeFlow w/ Sequence Augmentation | 6.62±0.35 | 6.68±0.52 | 5.03±0.47 | 41.10±0.29 | 40.87±0.35 | 51.59±0.42 |

Table 1: We demonstrate the recovery ratio (accuracy) and perplexity averaged on three trials on CATH 4.2 different subsets. All baselines are reproduced by (Gao et al., 2022). * indicates that the results are reported on CATH 4.3. For probability flow models (e.g. Diffusion, Flow), the perplexity is estimated with the method proposed in neural ODE (Chen et al., 2018).

| Method | MeFlow | ESM-IF | ProteinMPNN | PiFold |
|---|---|---|---|---|
| Recovery Ratio (%) ↑ | **61.22** | 44.47 | 49.28 | 60.84 |
| RMSD (Å) ↓ | **1.64** | 1.83 | 1.70 | 1.66 |

Table 2: For 50 randomly picked single-chain proteins, we report the recovery ratio accuracy and RMSD. We generate the stucture with OpenFold (Ahdritz et al., 2022).

**Main Results on Quality Measurement ❶** As described in Table 1, our proposed method surpasses other state-of-the-art approaches, including ESM-IF (Hsu et al., 2022), PiFold (Gao et al., 2022), and ProteinMPNN (Dauparas et al., 2022), in terms of recovery ratio. ❷ *Compare with Classification Model:* The classification model is trained using cross-entropy loss and lacks random noise in the input. Throughout both training and inference, it takes the micro-environment as input and outputs the corresponding amino acid class. When we use the same model architecture, we notice that we

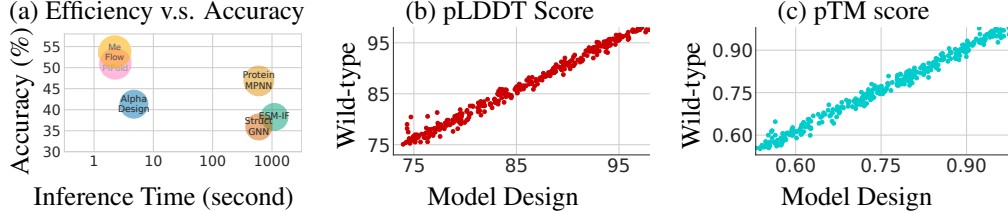

Figure 2: In (a), our MeFlow achieves the highest recovery ratio and lowest inference cost in the inverse folding task, as compared to other existing methods. The inference cost is measured on 100 amino acid sequences with an average length of approximately 1.6k on an NVIDIA A100. We do not compare with (Mao et al., 2023) since they train with larger datasets. In (b) and (c), we generate amino acid sequences, and run AlphaFoldV2 to calculate the pLDDT and pTM scores, respectively. We calculate 500 data points. Compared with wild-type sequences, our model's designed sequences show linearly-correlated in terms of Pearson correlation, similar pLDDT and pTM scores.

can achieve the highest recovery ratio, which outperforms all the baseline methods. ❸ *Compare with Diffusion Model:* Keep the same architecture, we use diffusion model to construct the learning objective. Using one step to inference, we notice that our flow model outperforms the baseline diffusion model, and we further compare the performance with different inference step later. ❹ *Efficiency v.s. Accuracy:* Our inference process is highly efficient, as it relies solely on atoms in the micro-environment. This enables faster inference speeds, especially for long protein sequences as shown in Figure 2(a). We observe that our method delivers optimal performance when balancing efficiency and accuracy. ❺ *AlphaFold Metrics:* In real protein design problems, a given backbone can come to multiple potential sequences, and our interest is to identify these candidate sequences. Therefore, only giving one target sequence and measuring the perplexity and recovery accuracy depending is not a good practice. Considering the importance of generating non-wildtype sequences in protein design, we adopt a valuable approach to further validate the quality of our model predictions by applying OpenFold (Jumper et al., 2021) to get the 3D structures for our self-design sequences. We first measure AlphaFold metrics, pLDDT and pTM. pLDDT represents the per-residue accuracy of the structure while pTM provides a measure of the error for the predicted structure in 3D. For a given protein structure, we predict the amino acid sequences based on the target fold Then, we feed the wild-type sequences and the predicted sequences into OpenFold and get the pLDDT and pTM scores. As demonstrated in Figure 2(b) and (c), the predicted sequences achieve comparable scores as the original ones. In terms of Pearson correlation, the difference between our method and PiFold is not significant.

❻ *Measure and Compare RMSD*: We conduct an additional experiment examining the RMSD values (specifically, RMSD of the Carbon alpha atomic coordinates) between the structure of ground truth sequence and the structure of our model output predicted by OpenFold. We randomly select 50 single-chain proteins (*e.g.,* 1qp2, 1f0m, 2wnm, 5jrt, 2qff), comparing the original OpenFold folding results and the folded results of model predicted sequence. As demonstrated in Table 2, we notice that our MeFlow can achieve better recovery ratio and RMSD (Root Mean Square Deviation) scores. We slightly improve the PiFold results while improve ESM-IF by a large margin (*e.g.*, recovery ratio is improved from $44.47\%$ to $61.22\%$, RMSD is improved from $1.82$Å to $1.64$Å).

| Method | MeFlow | MPNN (Beam Search) | MPNN (Temperature) | Pifold |
|---|---|---|---|---|
| Recovery Ratio (%) ↑ | **61.22** | 46.43 | 43.27 | 57.46 |
| RMSD (Å) ↓ | **1.64** | 1.73 | 1.78 | 1.68 |
| Relative Hamming Distance ↑ | **0.18** | 0.12 | **0.18** | 0.09 |
| Cosine Similarity ↓ | 0.85 | 0.88 | **0.83** | 0.92 |

Table 3: Here, we apply all the methods to 50 backbone structures, generate 10 protein sequences for each case, and then pass all sequences through ESM to get hidden representations for calculating cosine similarity. MPNN denotes ProteinMPNN.

**Evaluate Diversity** *Setting:* As demonstrated in Table 3, we evaluate the diversity of different method. To assess diversity, we first consider the relative Hamming distance since there is no insertion or deletion operation in the design space. In addition to sequence space, we also aim to evaluate diversity within a meaningful hidden space. we measure the similarity with the pretrained ESM2 (Lin et al., 2022) latent space.

| Recovery Ratio | 100-Step | 10-Step | 1-Step | Reflow 1-step |
|---|---|---|---|---|
| Diffusion | 53.34±0.42 | 52.23±0.48 | 50.52±0.44 | - |
| MeFlow | 53.68±0.48 | 53.56±0.52 | 53.25±0.46 | 53.53±0.45 |

Table 4: We showcase the recovery ratio (accuracy) on CATH using varying step models for both MeFlow and the diffusion model. Leveraging *ReFlow*, we achieve a reduction in inference time, while preserving the recovery ratio at levels comparable to multi-step inference approaches.

*Baseline Setup:* In autoregressive models, beam search can generate diverse candidate sequences. Adjusting the softmax temperature and implementing random sampling can also introduce diversity; we employ a temperature setting and we set $T = 2$ to do random sampling. On the other hand, non-autoregressive models like PiFold can introduce diversity through inference-time dropout. In our experiments, we benchmark ProteinMPNN using two strategies: a beam search with a size of 10 and random sampling. For PiFold, we apply inference-time dropout with $p = 0.1$.

*Results:* As demonstrated in Table 3, our MeFlow achieves the best trade-off between diversity and quality. MeFlow consistently outperforms all the baselines in terms of achieving a favorable balance between quality and diversity. Besides, compared to beam search, temperature softmax and random sampling can get better diversity. While MeFlow has similar diveristy socres as temperature softmax, we get better recovery ratio and RMSD.

**Compare with Diffusion Model** Our observations in Table 4 indicate that, in our problem settings, reducing the number of flow steps to one does not compromise accuracy for our MeFlow. Contrarily, the performance of the diffusion model significantly diminishes upon reducing the inference step. While we use enough number of steps to do inference, these two models achieve almost the same performance. Notably, we notice that employing a one-step model with *ReFlow* leads to a slightly higher recovery ratio (53.53%) compared to directly apply one time step (53.25%). In conclusion, MeFlow consistently surpasses the diffusion model across all the settings with different time steps.

**Evaluate on Interface Dataset** Beyond the widely-used CATH benchmark, we have curated our own dataset and present results across multiple domains. When trained on the *Interface* dataset and evaluated on various domain test sets, MeFlow consistently exhibits robust performance on diverse protein interfaces, as illustrated in Table 5. We detail the recovery ratios for distinct interface scenarios; for instance, 'RNAs' reveals the recovery ratio specific to the protein-RNA interface. Subsequent experiments further affirm that by leveraging the interface dataset, we can achieve enhanced capabilities in zero-shot protein engineering.

## 4.2  PROTEIN ENGINEERING TASKS (ZERO-SHOT ΔΔG PREDICTION)

**Experiment Settings** To further verify the ability of protein engineering, ① we apply zero-shot ΔΔG prediction on FireProtDB (Stourac et al., 2021) cleaned by Chen et al. and two other commonly used benchmark, P53 (Danziger et al., 2009) and Myolobin (Montanucci et al., 2019). The problem is predicting whether 'if a mutation can potentially improve a protein phenotyp. Following literature (Lin et al., 2022; Notin et al., 2022), we apply $\log(p_{\mathrm{mut}}/p_{\mathrm{wt}})$ as the zero-shot score to predict whether a mutation is good. $p_{\mathrm{mut}}$ stands for the probability of the mutated amino acid, and $p_{\mathrm{wt}}$ represents the probability of wild-type amino acid. ① To verify the generalization, we use two different 3D structure files for P53, one is the most commonly-used structure with pdb code *2ocj* and the other is *3q05* which has the binded DNA structure. ③ We evaluate our model using both Pearson and Spearman correlations. Additionally, we employ a range of other classification and regression metrics to evaluate our model's enhancements.

**Results** We predict the change in protein stability (i.e., ΔΔG) upon amino acid substitutions without requiring any experimental measurements or prior training on specific mutations. ① Table 6 shows the performance of different models on zero-shot ΔΔG prediction on several different benchmarks.

| | AEMetal | CarbonHyprate | Metal | RNAs | AlkaiMetal | Halogen | TransitionMetal | Antigen | Ligand |
|---|---|---|---|---|---|---|---|---|---|
| #Amino Acids (k) | 3.6 | 0.8 | 7.5 | 1.9 | 1.5 | 3.0 | 3.9 | 0.4 | 28.6 |
| MeFlow | 67.3±0.3 | 64.0±0.3 | 69.5±0.3 | 61.3±0.3 | 59.4±0.3 | 54.4±0.3 | 72.9±0.3 | 49.0±0.3 | 54.8±0.2 |
| Classifier | 67.8±0.2 | 64.7±0.1 | 70.8±0.1 | 62.1±0.2 | 59.9±0.2 | 55.0±0.2 | 73.4±0.2 | 49.5±0.2 | 55.6±0.1 |

Table 5: We assess the performance (recovery ratio) of our model on multiple test domains. '#Amino Acids' denotes the number of micro-environments.

| Dataset | FireProtDB | | P53 (2ocj) | | P53 (3q05) | | Myoglobin | |
|---|---|---|---|---|---|---|---|---|
| | Pearson ↑ | Spearman ↑ | Pearson ↑ | Spearman ↑ | Pearson ↑ | Spearman ↑ | Pearson ↑ | Spearman ↑ |
| ProteinMPNN | 0.24 | 0.28 | 0.35 | 0.40 | 0.37 | 0.42 | 0.35 | 0.35 |
| PiFold | 0.18 | 0.20 | 0.34 | 0.34 | 0.33 | 0.33 | 0.18 | 0.22 |
| ESM-1b | 0.22 | 0.25 | 0.35 | 0.37 | - | - | 0.28 | 0.30 |
| Stability Oracle | 0.58 | 0.62 | 0.73 | 0.62 | - | - | 0.62 | 0.62 |
| ThermoNet | - | - | 0.45 | - | - | - | 0.38 | - |
| Classifier | 0.24 | 0.27 | 0.34 | 0.37 | 0.33 | 0.37 | 0.32 | 0.32 |
| MeFlow | 0.27 | 0.30 | 0.38 | 0.40 | 0.45 | 0.42 | 0.47 | 0.41 |
| MeFlow w/ Dropout | 0.26 | 0.28 | 0.36 | 0.37 | 0.42 | 0.44 | 0.43 | 0.41 |
| MeFlow w/ Aug | 0.29 | 0.41 | 0.42 | 0.49 | 0.45 | 0.50 | 0.44 | 0.44 |
| MeFlow w/ Aug (Interface) | **0.31** | **0.33** | **0.45** | **0.55** | **0.51** | **0.59** | **0.48** | **0.47** |

Table 6: We assess the performance of different models on zero-shot fitness ($\Delta\Delta G$) prediction tasks. ThermoNet and Stability Oracle are supervised models , and the numbers are from their paper.

The table reports two evaluation metrics: spearman correlation coefficient and pearson correlation coefficient, to make a comprehensive evaluation. ② The first two rows of the table report the performance of two baseline models, ProteinMPNN and PiFold. We notice that, while PiFold surpasses ProteinMPNN in the recovery ratio, ProteinMPNN is a better model for zero-shot $\Delta\Delta G$ prediction. ③ We observe that by transforming the classifier into a generative model, termed MeFlow, there's an improvement across all metrics. While the classifier model exhibits a higher recovery ratio, it falls short as an effective zero-shot predictor for $\Delta\Delta G$. By introducing our proposed sequence augmentation and training on our Interface dataset, we notice further improvements on these correlation metrics. ④ In Figure 3, we further compare classifier model and our best generative model with more metrics. Remarkably, we observe substantial enhancements in both regression and classification metrics. For example, MeFlow improves the binary classification accuracy for $\Delta\Delta G$ (whether the mutation is stable or not) in all four datasets. ⑤ In summary, our observations reveal that while the classifier model surpasses MeFlow in recovery ratio (as seen in Table 1), PiFold exceeds ProteinMPNN in the same metric, PiFold outperforms ProteinMPNN in recovery ratio, MeFlow and PiFold get better metrics for zero-shot $\Delta\Delta G$ prediction tasks. Our MeFlow achieves the best results in zero-shot $\Delta\Delta G$ prediction.

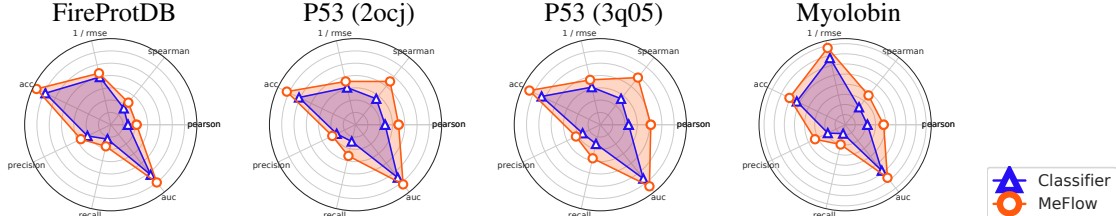

Figure 3: We compare MeFlow and the classifier model with the same architecture. For all the metrics, larger is better. Detailed numbers are shown in the appendix.

## 5 CONCLUSION

In this research, we present the MeFlow framework, tailored for protein engineering based on target folds. MeFlow stands out by offering enhanced computational efficiency and a superior ability to produce varied sequences. Our empirical findings underscore MeFlow's proficiency in generating an array of sequences tailored to a specific protein scaffold. From our investigations, two notable insights emerge: ❶ First, our model, when provided with local chemical environments instead of the complete backbone as inputs, can attain comparable or even superior recovery ratios. This revelation propels a deeper question: Can protein inverse folding challenges be proficiently addressed using just the local chemical environment? How can we effectively harness the benefits of the complete backbone? ❷ Secondly, when evaluating various models on zero-shot $\Delta\Delta G$ prediction, we find that achieving a higher recovery ratio doesn't invariably translate to superior zero-shot performance. This brings forth another contemplative query: Should inverse folding primarily focus on maximizing the recovery ratio? Moving forward, we plan to verify the relation between recovery ratio and protein engineering performance. Moreover, We foresee the potential to broaden the application of our flow-based generative model, aiming to address a wider range of conditional generation challenges within protein research, such as protein folding and multiple sequence alignment (MSA).

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

# A  ARCHITECTURE

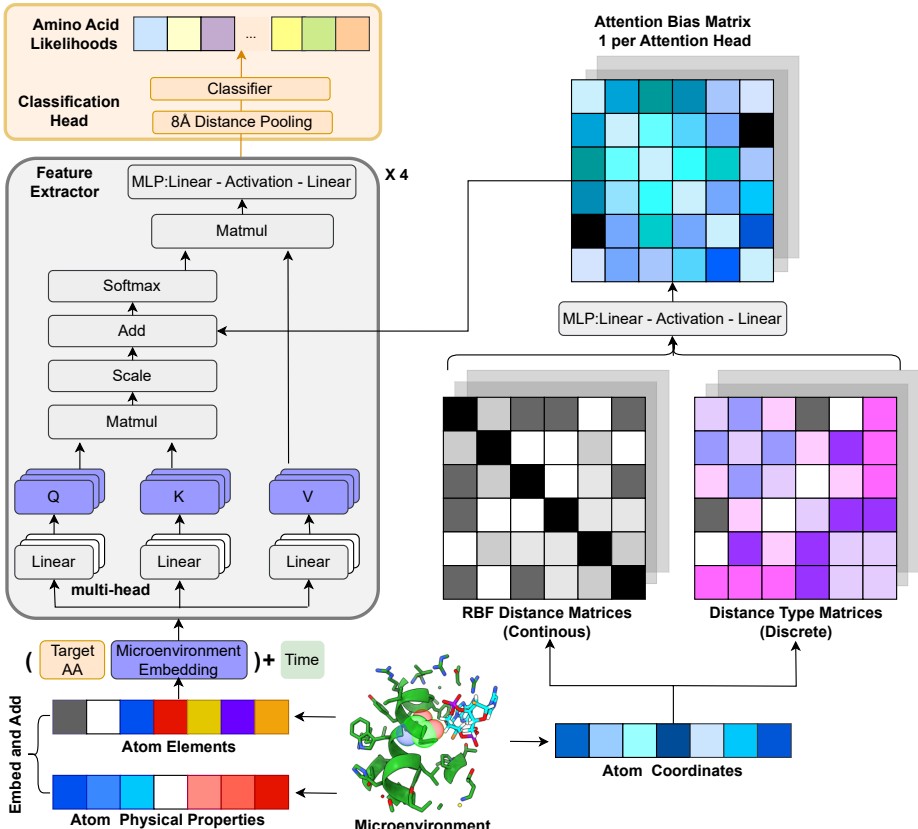

Figure 4:  We demonstrate the detailed neural network architecture used in this paper.

Our architecture is based on the graph transformer neural network architecture proposed in (Diaz et al., 2023) and we demonstrate the details in Figure 4. Specifically, we use this architecture to construct $v_\theta$, which is designed to capture the local environment surrounding the target amino acid. The graph transformer model takes this environment as input, allowing us to generate accurate predictions. Specifically, we define the Carbon-$\alpha$ as the center and extract all the atoms within a radius of $n\dot{A}$. The neural network takes in the coordinates, atom types, and physical properties of the atoms, and applies an embedding layer to convert the categorical atom types into continuous representations. Once we do not have the side chain information, we cannot have accurate physical properties, and therefore we only introduce little information in our model. Our included physical properties contain two channels, the partial charges and the surface or core information. The physical properties contain categorized partial charge (negative, neutral and positive) and whether the amino acid is on the surface (true or false). The embeddings of atom types and physical property types are concatenated to form the input features, which then pass through several attention blocks. In each attention block, there are two attention layers and one MLP layer. The attention layers use the atom-wise euclidean distance to calculate the attention bias, which is based on distance matrices and RBF functions. For masked amino acid predictions, we remove every atom in the target amino acids and predict the corresponding amino acid type. For inverse folding, we remove all the side chains in a micro-environment and only give the backbone atoms.

**Encoder-Decoder Architecture**   We also propose an Encoder-Decoder architecture as shown in Figure 5. We notice that it takes the encoder-decoder architecture a longer time to train while the final results cannot outperform our current architecture.

**Impact of Physical Properties**   In this work, except for atom types and coordinates, we also introduce physical properties as the input feature. We use 1) three-class charge information, negative,

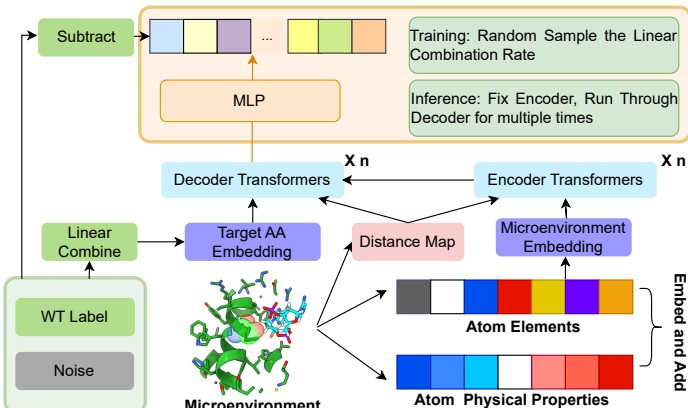

Figure 5: The alternative architecture: encoder-decoder architecture, in which we create three different distance map for encoder-only, decoder-only and encoder-decoder attentions. During inference, the input to the decoder is pure random noise, and we pass the decoder model multiple times to get the output amino acids.

neutral, or positive; 2) two-class surface information, on the surface or in the core. In practice, we notice that these features help us to train the model faster. Without these features, we can still achieve a similar recovery ratio. As shown in Table 7, with longer training time, without physical property model finally achieves a similar recovery ratio.

|  | #Iterations | Recovery Ratio |
|---|---|---|
| w/ pp | 300K | 53.53±0.45 |
| w/ pp | 1000K | 53.75±0.42 |
| w/o pp | 300K | 45.88±0.44 |
| w/o pp | 1000K | 48.93±0.49 |

Table 7: We show that physical properties help us get similar results with fewer training cost.

## B  EXPERIMENT SETTINGS

**AlphaFold Scores**   We report pLDDT and pTM scores in the experiments. To calculate these scores, we run five models and average their output pLDDT and pTM to get the score for one input sequence. Both the pLDDT and pTM scores provide an assessment of the quality and accuracy of the predicted protein structure by AlphaFold. Higher scores indicate better predictions, while lower scores indicate lower confidence in the predicted structures.

**ESM-2 Embedded Scores**   We convert the protein sequences into hidden representations using the ESM-2 model. We then extract the hidden representations (embeddings) of the target amino acid sequences from the ESM-2 model and then do average pooling. These hidden representations encode the sequence information and capture important characteristics of the proteins. Finally, we apply cosine similarity and provide a quantitative measure of how similar or dissimilar the protein sequences are.

**Zero-shot $\Delta\Delta$G Prediction**   To calculate the zero-shot $\Delta\Delta$G, we get the $p_{\mathrm{mut}}$ and $p_{\mathrm{wt}}$ from our model. In practice, for diffusion and flow generative models, we random sample 100 noise, average the outputs, and apply softmax to get the proability. Then, we calculate the log ratio score $=\log\frac{p_{\mathrm{mut}}}{p_{\mathrm{wt}}}$. Once score $> 0$, it indicates that this is a stabilizing mutation ($\Delta\Delta$G ¡ 0).

For the FireProt dataset, We follow (Chen et al.), and filter out neutral mutations where $|\Delta\Delta G| < 0.5$. We also filter out data points with standard deviation $> 0.25$ who have been measured by multiple times while the results are very diverse. Finally, we get $1,721$ data points. For the other datasets, we follow (Diaz et al., 2023) to process the data.

## C    WEAKNESSES

We recognize the disparity between machine learning-based sequence generation and the tangible processes involved in real protein engineering within wet labs. To enhance clarity, we underscore the necessity for additional steps dedicated to facilitating the seamless transition from in silico predictions to practical implementation in wet-lab settings.

