# OpenReview forum: "Microenvironment Probability Flows as Proficient Protein Engineers"
_ICLR.cc/2024/Conference — Submitted to ICLR 2024_

### Official Review · Reviewer_XuyZ · 2023-10-28

**Soundness:** 3 good
**Presentation:** 3 good
**Contribution:** 4 excellent
**Rating:** 6
**Confidence:** 3

**Summary:**

The manuscript introduces an inverse folding method based on probability flow models, featuring data augmentation with sequences bearing high similarity to enhance prediction diversity. The manuscript also contends that leveraging the microenvironment during the inverse folding process yields performance comparable to the use of global structural information.

**Strengths:**

1. Since only the microenvironment is utilized in the model, the proposed method offers significantly faster inference compared to existing methods while maintaining performance comparable to state-of-the-art methods.

2. The experiments are comprehensive, assessing the performance of the proposed method from various perspectives.

**Weaknesses:**

1. The pLDDT score is designed to evaluate the accuracy of structure predictions generated by AlphaFold. Using this metric to assess the authenticity of generated sequences and concluding that 'the model is likely to work in the real world' is logically inconsistent.

2. There are come typos in the manuscript.

**Questions:**

When constructing the microenvironment dataset, why was random sampling used to backfill up to 200 residues instead of employing distance measures for sampling?

---

> ### Author Response · Authors · 2023-11-20
> **Response to Reviewer XuyZ**
>
> **The pLDDT score is designed to evaluate the accuracy of structure predictions generated by AlphaFold. Using this metric to assess the authenticity of generated sequences and concluding that 'the model is likely to work in the real world' is logically inconsistent.**
>
> Thank you for your thoughtful comments. Our choice to utilize pLDDT and RMSE in assessing the generated sequences is motivated by two key considerations. Firstly, the decision is aligned with practices found in relevant literature, where one or both of these metrics have been commonly employed.
> Secondly, in the context of evaluating inverse folding or engineering models, researchers are particularly interested in ensuring the 'consistency' between structure-to-sequence and sequence-to-structure predictions. Given AlphaFold's proficiency in predicting protein structures, leveraging it to assess this consistency appears logical and is in line with the objectives of our study.
>
> **There are come typos in the manuscript.**
>
> Thanks for your comments. In the revision pdf, we have addressed typos mentioned by other reviewers.
>
>
> **When constructing the microenvironment dataset, why was random sampling used to backfill up to 200 residues instead of employing distance measures for sampling?**
>
> Thank you for your insightful suggestions. We employed a 200 residues threshold to prevent oversampling in the context of long protein sequences. Considering your suggestion and the importance of maintaining a balanced dataset, we recognize the value of incorporating distance measures to achieve a more nuanced understanding of local chemical environments. We view this approach as a promising alternative for future work in our data engineering efforts.

---

### Official Review · Reviewer_VmMV · 2023-10-30

**Soundness:** 3 good
**Presentation:** 1 poor
**Contribution:** 2 fair
**Rating:** 5
**Confidence:** 3

**Summary:**

This paper addresses the Inverse Protein design problem, where the goal is to predict the amino acid sequence corresponding to a given 3D protein backbone. In contrast to previous approaches that rely on the entire protein backbone, this work introduces three key contributions. First, it employs a generative flow network to generate each amino acid based solely on the local environment information of its intended structural placement. During inference, their model only requires one step of integration resulting in a significant speed improvement. Additionally, through data augmentation, the model demonstrates the ability to provide a more diverse set of sequences compared to state-of-the-art methods. Finally, the paper showcases the unique capability of their generative flow network to perform zero-shot single-point mutation fitness prediction, a feat not easily achievable with other methods

**Strengths:**

Originality:
The primary source of innovation lies in the utilization of rectified flow in conjunction with microenvironment data to enhance both speed and data diversity in the model. Of particular interest is the zero-shot experiment; however, it remains somewhat challenging to discern whether any additional modeling decisions influenced its success, or if it can be primarily attributed to the adoption of the rectified flow model.

Quality:
The experiments conducted in this study are comprehensive and serve to highlight the method strengths: speed, data diversity, zero-shot stability prediction, and robustness when varying the number of integration/diffusion steps.

**Weaknesses:**

Clarity:
Substantial effort is required to refine the content for improved clarity. For example, there is a need for a more detailed explanation of the classifier in Table 6.

This work involves the application of the rectified flow network to the domain of inverse protein design. This choice yields a unique capability in zero-shot stability prediction. However, when examining the overall performance and speed enhancements, they appear relatively minor when compared to certain state-of-the-art methods like PiFold.

The utilization of data augmentation contributes to increased sequence diversity, yet it is worth noting that this technique could be similarly applied to other methods to achieve comparable diversity benefits.

**Questions:**

Suggestions:
Improve the writing and submit the work to a more applied venue since the ML contribution is very limited.

---

> ### Author Response · Authors · 2023-11-20
> **Response to Reviewer VmMV**
>
> **Clarity: Substantial effort is required to refine the content for improved clarity. For example, there is a need for a more detailed explanation of the classifier in Table 6.**
>
> Thank you for your feedback. As we mentioned in experiment settings of section 4.2, this is a zero-shot classifier built with log(p_mut / p_wt), which is a common practice in literature (Lin et al., 2022; Notin et al., 2022).
>
> **This work involves the application of the rectified flow network to the domain of inverse protein design. This choice yields a unique capability in zero-shot stability prediction. However, when examining the overall performance and speed enhancements, they appear relatively minor when compared to certain state-of-the-art methods like PiFold.**
>
> Thanks for your comments.
> First, It's important to note that the inverse folding problems inherently lack side chain information in the structure. Consequently, the achievable recovery ratio is inherently limited in this context.
> Secondly, we hold the view that the recovery ratio may not be an optimal metric in practical applications. This metric considers only one amino acid as the correct answer for each position. However, in protein engineering practice, multiple amino acids can be valid at a given position, aligning with the diversity inherent in the field. We are actively exploring ways to establish more appropriate metrics and benchmarks to comprehensively evaluate all published models.
>
> **The utilization of data augmentation contributes to increased sequence diversity, yet it is worth noting that this technique could be similarly applied to other methods to achieve comparable diversity benefits.**
>
> From our perspective, the data augmentation is better than other data augmentation methods, e.g. label smoothing, dropout, etc. The sequence alignment augmentation contains more biology information, and therefore, as demonstrated in Table 6, it improves the zero-shot regressor and classifier by a large margin. We further display the dropout baselines in the revision and show that it cannot contribute. While this method can be applied to the other prior arts. We propose it here so in the future other methods can use it to improve their performance. We will make the generalization of our data augmentation technique more clear in the writing.
>
> **Suggestions: Improve the writing and submit the work to a more applied venue since the ML contribution is very limited.**
>
> Thank you for your feedback. We have carefully addressed the typos mentioned by other reviewers in the revised version of the document. For the machine learning contributions, from our perspective, in this work we make the following contributions:
>
> A Simplified Approach for Discrete Outputs: Our empirical findings demonstrate that for discrete outputs, the conventional design of discrete noise [1, 2] or the addition of Gaussian noise to the embedding space of discrete data [3] is unnecessary. We show that generating one-hot encoding without altering the algorithm is sufficient. This is an advantage given by the flow model that we demonstrate works in proteins.
>
> Biology-based Data Augmentation: We propose to do data augmentation with sequence alignment information. For the protein problems, validated experimental structures are limited (20k~30k). Therefore, data augmentation is an interesting problem and a contribution to the protein machine learning community.
>
> Performance Improvement in Practice: In practical applications, we have observed performance enhancements, as detailed in the paper, that may be used by others in the protein machine learning community.
>
> To enhance clarity, we will emphasize these key points in the manuscript, ensuring a more straightforward understanding of our contributions once we have the extra page.  We appreciate your attention to these aspects and believe that these refinements will strengthen the overall quality of our work.
>
> [1] Structured denoising diffusion models in discrete state-spaces
>
> [2] Argmax flows and multinomial diffusion: Learning categorical distributions
>
> [3] Diffuseq: Sequence to sequence text generation with diffusion models
>
> [4] Diffusion-LM improves controllable text generation.

---

### Official Review · Reviewer_1cV9 · 2023-10-31

**Soundness:** 2 fair
**Presentation:** 2 fair
**Contribution:** 2 fair
**Rating:** 5
**Confidence:** 3

**Summary:**

The authors create a probability flow model to predict the amino acid from the 3D microenvironment around the given amino acid.

**Strengths:**

Generally, I think this is an interesting and tractible approach.

I think the data augmentation process makes sense. I think the authors could likely be even more aggressive with this approach (more sequence diversity).

I think the speed and efficiency of this approach is good.

I think the comparison across multiple tasks is good to see where this method shines.

**Weaknesses:**

As someone that has undergone an actual wet-lab protein engineering campaign, the tools presented here should be defined as annotation or sequence generation. There are not clear steps to move these entities out of in silico predictions.

It would be great to define “recovery ratio” before using it a number of times in the work. Ideally the reader doesn’t have to hunt through a number of references to find its definition. This is also less familiar than the pLDDT or pTM scores.

“Compare with Classification Model: When we use the same architecture and input, but train a classifier instead of a generative model…” What is the classifier? What are you training it on? This should be clearly defined to the reader. Coupled with no definition of the “recovery ratio”, this section is very hard to follow.

For all the tables (including Table 1), it’d be great to have up or down arrows if a “good” metric should be large or small, respectively.

The cosine similarity results in Table 3 are difficult to make conclusions from. What are meaningful, statistically significant distances for cosine similarity? Each of these models have quite different representations and architectures, many of which haven’t been designed for this task.

“Experiment Settings To further verify the ability of protein engineering, we apply zero-shot ∆∆G prediction on FireProtDB….” This is sequence annotation.

There are many different zero-shot mutation effect predictors, one of which is in the ESM family of models. I would like to see that baseline in section 4.2.

Nit: “and” is spelled wrong a number of times in the manuscript, sometimes as “nad” or “adn”.

**Questions:**

“As demonstrated in Table 3, we evaluate the diversity of different method. To assess diversity, we first consider the relative Hamming distance since there is no insertion or deletion operation in the design space.” - What about Levenshtein Distance? This is pretty common to deal with indels.

“Hence, we propose that differences within this hidden space could represent a ‘hidden’ dimension for important concepts, such as MSA, and therefore applying cosine similarity to measure diveristy in the ESM2 latent space.” I don’t think I fully follow or understand here.

“we apply log(pmut/pwt) as the zero-shot score to predict whether a mutation is good.” What exactly do you mean by “good”? I would prefer if the phrasing was “if a mutation affects function”.

“In real protein design problems, a given backbone can come to multiple potential sequences, and our interest is to identify these candidate sequences.” What does this mean? If this is a real problem biologists can encounter, can you provide and example or citation?

I see in this work that ESM is used for creating embeddings and some of the other validations. Was this version of ESM blinded to CATH as well, as your splits have been done in model creation?

In the mutation effect prediction task, can this method predict the effect of multiple mutations?

Does sequence augmentation actually help with your method? If not, why include it?

How well does the “Classification” model in Table 1 compare in all the other experiments below?

While I appreciate the number of metrics in Figure 3, I think it is a bit overkill. Also, the 1/rmse for Myolobin is off the chart.

---

> ### Author Response · Authors · 2023-11-20
> **Response to Reviewer 1cV9**
>
> **As someone that has undergone an actual wet-lab protein engineering campaign, the tools presented here should be defined as annotation or sequence generation. There are not clear steps to move these entities out of in silico predictions.**
>
> Thank you for your comments, we emphasize this in the revision appendix “We recognize the disparity between machine learning-based sequence generation and experimental validation of computationally generated protein sequences. To address your question, we first develop methods, iteratively improve them, and submit them to machine learning conferences with the intent of sharing our in silico experimental result with the machine learning community. Once we feel our methods cannot be improved further in silico, we collaborate with experimental protein engineers to experimentally validate our methods. However, these experimental collaboration projects get submitted to protein journals such as Nature, Cell, ACS, journals rather than machine learning conferences.
> We will move the discussions to the main text once we have an extra page.
>
> **It would be great to define “recovery ratio” before using it a number of times in the work. Ideally the reader doesn’t have to hunt through a number of references to find its definition. This is also less familiar than the pLDDT or pTM scores.**
>
> Thank you for your comments, we highlight the recovery ratio refers to the wildtype accuracy in the revision.
>
> **“Compare with Classification Model: When we use the same architecture and input, but train a classifier instead of a generative model…” What is the classifier? What are you training it on? This should be clearly defined to the reader. Coupled with no definition of the “recovery ratio”, this section is very hard to follow.**
>
>  The classification model is trained using cross-entropy loss against the wildtype label and lacks adding random noise to the input. Throughout both training and inference, it takes the backbone micro-environment as input and outputs the corresponding wildtype amino acid class. In contrast, the flow model necessitates a linear combination of the target and random noise as input during training. During inference, it first starts with random noise and the flow process uses the backbone microenvironment to denoise it into the predicted amino acid class. Additional details addressing these distinctions have been incorporated in the revised version.
>
> **For all the tables (including Table 1), it’d be great to have up or down arrows if a “good” metric should be large or small, respectively.**
>
> Thank you for your comments, we add these arrows in the updated version.
>
> **The cosine similarity results in Table 3 are difficult to make conclusions from. What are meaningful, statistically significant distances for cosine similarity? Each of these models have quite different representations and architectures, many of which haven’t been designed for this task.**
>
> Thank you for bringing this confusion to our attention. In Table 3, we generate sequences for 50 randomly selected proteins/backbones. For each of these backbone scaffolds, we run MPNN, PiFold, and MeFlow to generate 10 sequences, giving us a total of 500 sequences for each model. For the 10 sequences generated, we embed them with ESM2 and measure their cosine similarity in the ESM2 latent space. Since the sequences are being compared in solely ESM2 latent space, the cosine similarity of the sequence’s embedding for a given scaffold provides a fair comparison and a meaningful metric. We do this for each of the 50 randomly selected scaffolds with MPNN, PiFold, and MeFlow separately.
> We will clarify this in the Table caption and paper to make this more clear once we have extra page.
>
> **“Experiment Settings To further verify the ability of protein engineering, we apply zero-shot ∆∆G prediction on FireProtDB….” This is sequence annotation.**
>
> Thank you for your comment. Yes, ∆G is sequence annotation and ∆∆G is an annotation for a point mutation calculated by given ∆G of two sequences. We use these annotations to evaluate how well our model correlates with them.  In our context, ∆∆G represents the change free energy of folding based on a point mutation.
>
>
>
> **There are many different zero-shot mutation effect predictors, one of which is in the ESM family of models. I would like to see that baseline in section 4.2.**
>
> Thank you for your comments, we list some supervised models and zero-shot models for the comparison. We present ESM-1b, ThermoNet and Stability Oracle in Table 6. ThermoNet is a well-known supervised trained ddG predictor, while Stability Oracle is a recently published SOTA model. We demonstrate that our zero-shot performance is even better than the supervised ThermoNet model.
>
> **Nit: “and” is spelled wrong a number of times in the manuscript, sometimes as “nad” or “adn”.**
>
> Thank you for your comments, we fix these typos.

---

> ### Author Response · Authors · 2023-11-20
> **Reponse to Reviewer 1cV9 [2]**
>
> **“As demonstrated in Table 3, we evaluate the diversity of different method. To assess diversity, we first consider the relative Hamming distance since there is no insertion or deletion operation in the design space.” - What about Levenshtein Distance? This is pretty common to deal with indels.**
>
> Thank you for your comments. In our problem set, the sequence length is fixed (no insertions and deletions), and the position in the sequence corresponds to the position in the structure. In this case, there is no distinction  between Levenshtein Distance and Hamming distance and they can be considered equal.
>
> **“Hence, we propose that differences within this hidden space could represent a ‘hidden’ dimension for important concepts, such as MSA, and therefore applying cosine similarity to measure diversity in the ESM2 latent space.” I don’t think I fully follow or understand here.**
>
> Thanks for the comments, we simplify this to ‘we measure the similarity with the pretrained ESM2 latent space’. Our key insight is based on the fact that ESM2 replaced the need of explicit MSAs in ESMFold. This enabled us to hypothesize that its latent space must  encompass rich information about MSAs.
>
> **“we apply log(pmut/pwt) as the zero-shot score to predict whether a mutation is good.” What exactly do you mean by “good”? I would prefer if the phrasing was “if a mutation affects function”.**
>
> Thanks for the comments, ‘good’ is confusing and we rewrite this to ‘if a mutation can potentially improve a protein phenotype, such as stability (∆∆G)”.
>
> **“In real protein design problems, a given backbone can come to multiple potential sequences, and our interest is to identify these candidate sequences.” What does this mean? If this is a real problem biologists can encounter, can you provide and example or citation?**
>
> As we mentioned in the third paragraph in the introduction section, It is quite common in nature for proteins that have a sequence similarity of at least ∼30% to have a structurally similar scaffold (Stern, 2013; Rost, 1999). Once we go below 30% sequence similarity, this fact begins to break down and is known as the “twilight zone” (Rost, 1999). This scaffold degeneracy is the underlying principle that drives the grouping of proteins into families and domains (e.g., Fox et al., 2014; Sillitoe et al., 2021; Paysan-Lafosse et al., 2023).
>
> We refer the readers and the reviewers to all these related works we mentioned, for the background and related information. We will add a citation to this statement to help the reader.
>
> **I see in this work that ESM is used for creating embeddings and some of the other validations. Was this version of ESM blinded to CATH as well, as your splits have been done in model creation?**
>
> Thanks for your valuable insight. To clarify, ESM is used as one way to measure the similarity in the hidden space. The ESM is trained on sequence similarity splitting while CATH is structure splitting, therefore, the ESM is not blind to the CATH sequences. We highlight this in the revision. However, we do not see this resulting in any issues since we are not training or evaluating the performance of ESM2.
>
> **In the mutation effect prediction task, can this method predict the effect of multiple mutations?**
>
> As mentioned in the experiment settings of section 4.2, we construct zero-shot predictors based on the flow model. For the multiple mutations, we can get a zero-shot model by adding single-mutation effects as demonstrated in [1] and ESM-1b paper. However, this naive addition does not take into account epistatic interactions. To get an accurate multiple-mutation predictor, it’s necessary to do supervised training with multiple mutation data, which is not the key focus of this paper.
>
> [1] Predicting a Protein’s Stability under a Million Mutations
>
> **Does sequence augmentation actually help with your method? If not, why include it?**
>
> As we mentioned in Table 6, the augmentation improves the zero-shot regressor performance by a large margin (for example, on FireProtDB, the spearman is improved from 0.30 to 0.41). Considering the literature, it even out-performs the supervised models, e.g., [1]. We list these numbers for further comparison in revision, and therefore we think our augmentation helps the model.
>
> [1]  Predicting changes in protein thermodynamic stability upon point mutation with deep 3D convolutional neural networks

---

> > ### Author Response · Authors · 2023-11-20
> > **Reponse to Reviewer 1cV9 [3]**
> >
> > **How well does the “Classification” model in Table 1 compare in all the other experiments below?**
> >
> > Thanks for your suggestion, we include it in several tables in the updated pdf, where it makes sense. Table 3 is about measuring the diversity while the classification model cannot directly generate diverse outputs. Table 4 is about sensitivity analysis about generation related hyper-parameters and has nothing to do with classification. In Table 5, we add the classifier results in the revision. In Table 6, we already demonstrate the classification model’s  zero-shot performance.
> >
> > **While I appreciate the number of metrics in Figure 3, I think it is a bit overkill. Also, the 1/rmse for Myolobin is off the chart.**
> >
> > Thanks for your comments, as we demonstrated in the caption, all the detailed numbers are in the appendix. We also update figure for Myolobin.

---

### Official Review · Reviewer_UWpB · 2023-10-31

**Soundness:** 2 fair
**Presentation:** 2 fair
**Contribution:** 2 fair
**Rating:** 5
**Confidence:** 4

**Summary:**

The authors introduce MeFlow, a diffusion model for inverse design of protein sequences from local atomic environments. They contrast MeFlow with inverse folding methods like ProteinMPNN and ESM-IF, which condition on global backbone structure information to generate sequences. They show that models conditioned on microenvironments achieve similar results, and they propose a data augmentation strategy based on sequence similarity. They show results on CATH, FireProtDB, and a newly introduced Interface dataset.

**Strengths:**

The work shows a complementary approach for inverse protein design that works about as well as inverse folding approaches that condition on global backbone structure. The method is evaluated in three different settings, going well beyond the usual evaluation for inverse design methods. Using sequence augmentation to improve diversity is an interesting idea.

**Weaknesses:**

There are some basic issues of clarity that make it difficult to interpret the results, particularly for the newly introduced tasks. The central claim that conditional generation using microenvironments allows for designing more diverse sequences that recover a target backbone structure is not sufficiently supported. The sequence augmentation strategy is underexplored.

**Questions:**

1. What atoms comprise the local environment? Are these all the backbone atoms?
2. The authors claim that their overarching goal is to explore conditioning on local atomic environments rather than global backbone structure for inverse design; the usual sequence recovery and diversity metrics do not show much difference between these paradigms. Have the authors considered tasks more related to remote homology detection?
3. Data augmentation via Uniref100 is interesting but underexplored. Do the authors have ideas or additional experiments that could improve the results using this style of sequence augmentation?
4. What physical properties are used to characterize the atoms?
5. What are the correlation coefficients in Fig 2b and c? Is this similar to other methods? What references or evidence suggest that achieving comparable pLDDT and pTM scores indicate that a model will “work in the real world”?
6. What baselines are relevant for Table 5?
7. Is the Interface dataset publicly available?
8. It may improve clarity to give names to the Losses in equations 4 and 7 to refer to them throughout the paper.
9. Figure 3 is visually interesting but not particularly informative and largely redundant with Table 6.

---

> ### Author Response · Authors · 2023-11-20
> **Reponse to Reviewer UWpB**
>
> **Weaknesses**
>
> Thanks for your comments, here we explain our innovations.
> We do not introduce new tasks. In Table 1, we compare recovery ratio which is commonly used in inverse folding models (e.g., PiFold). In Table 6, we compare zero-shot transferability, which is also commonly used in literature (e.g. ESM [1]).  The introduced Interface dataset is for better performance on the zero-shot task.
> We do not agree with the point “The central claim that conditional generation using microenvironments allows for designing more diverse sequences that recover a target backbone structure is not sufficiently supported”. In literature, researchers either compare recovery ratio (e.g., PiFold), or compare zero-shot single-mutation predictions (e.g., ESM). We do both settings, and demonstrate that our method is comparable or even better.
> For Data Augmentation: We propose to do data augmentation with sequence alignment information. For the protein problems, validated experimental structures are limited (20k~30k). Therefore, data augmentation is an interesting problem and a contribution, and we do not notice related works and therefore do not set up more comparisons.
> [1] Language models enable zero-shot prediction of the effects of mutations on protein function
>
>
> **What atoms comprise the local environment? Are these all the backbone atoms?**
>
> The microenvironment consists of the backbone atoms that are within 20Å of the Ca atom of the center residue. We will make edits in the language to make the definition of the inverse folding microenvironment more clear. We only task the backbone atoms to build the local chemical environment.
>
> **The authors claim that their overarching goal is to explore conditioning on local atomic environments rather than global backbone structure for inverse design; the usual sequence recovery and diversity metrics do not show much difference between these paradigms. Have the authors considered tasks more related to remote homology detection?**
>
> Thanks for valuable comments. We think an inverse folding model can be used as a zero-shot remote homology detector. However we have yet to run these experiments ourselves.
>
> **Data augmentation via Uniref100 is interesting but underexplored. Do the authors have ideas or additional experiments that could improve the results using this style of sequence augmentation?**
>
> Thanks for valuable comments. We think generating augmentations with sequence similarity is a new and not-explored idea. We know similar sequences can come to similar scaffolds and we plan to do more future works on this topic.
>
> **What physical properties are used to characterize the atoms?**
>
> As we mentioned in the appendix, Our included physical properties contain two channels, the partial charges and the surface or core information. The physical properties contain categorized partial charge (negative, neutral and positive) and whether the atom is buried.
>
> **What are the correlation coefficients in Fig 2b and c? Is this similar to other methods? What references or evidence suggest that achieving comparable pLDDT and pTM scores indicate that a model will “work in the real world”?**
>
> We update the manuscript and mention this is the Pearson correlation. We also add comments “In terms of Pearson correlation, the difference between our method and PiFold is not significant” in the updated pdf. We use pLDDT since ProteinMPNN [1] evaluate relative AlphaFold success rates in Figure 2. We notice this is not accurate and therefore remove this sentence “work in the real world” in the updated pdf.
>
> [1] Robust deep learning–based protein sequence design using ProteinMPNN
>
> **What baselines are relevant for Table 5?**
>
> Thanks for your comments, we further provide a classification model baseline in the revision. We mainly do the comparison in table 6, which shows training on our dataset, we get a better zero-shot model.
>
> **Is the Interface dataset publicly available?**
>
> The dataset is not publicly available yet, we will make the dataset we used available after acceptance.
>
> **It may improve clarity to give names to the Losses in equations 4 and 7 to refer to them throughout the paper.**
>
> Thanks for your comments, we will modify this once we have more space for the main text.
>
> **Figure 3 is visually interesting but not particularly informative and largely redundant with Table 6.**
>
> In Figure 3, we want to show more metrics to compare these two methods. Table 6 only shows two correlation metrics. Therefore, we do not think Figure 3 is redundant. A model with high correlation does not always refer to a model with high recall/precision.

---

> > ### Comment · Reviewer_UWpB · 2023-11-21
> >
> > I have the read the author's response and appreciate their rebuttal. Along with some of the other reviewers, my concerns related to the clarity of the work and the claims remain unchanged.

---

> > > ### Author Response · Authors · 2023-11-21
> > > **Official Response to Reviewer UWpB**
> > >
> > > Thank you for your valuable feedback and insightful suggestions again. Your comments have greatly contributed to the improvement of our work.

---

### Meta-Review · Area_Chair_S2ND · 2023-12-05

**Metareview:**

The manuscript introduces MeFlow, a probabilistic flow framework aimed at the inverse design of protein sequences using local atomic environments. This method contrasts with inverse folding techniques such as ProteinMPNN and ESM-IF, which rely on global backbone structure information for sequence generation. MeFlow demonstrates comparable results by conditioning on micro-environments and incorporates a data augmentation strategy based on sequence similarity. The efficacy of MeFlow is showcased through experiments on CATH, FireProtDB, and a newly introduced Interface dataset. Strengths of the study include the sensible data augmentation process, with potential for further enhancement through increased sequence diversity. The method's speed and efficiency are noted as positive aspects.

**Justification For Why Not Higher Score:**

The manuscript is critiqued for its subpar clarity. The experimental results presented are not compelling enough to conclusively demonstrate the practical utility of the method. The advancements it offers compared to the state of the art are seen as minimal. Furthermore, the sequence augmentation strategy, while promising, is considered underexplored and could benefit from further development and analysis.

**Justification For Why Not Lower Score:**

N/A

---

### Decision · Program_Chairs · 2024-01-16

Reject